# Polycyclic Aromatic Hydrocarbons in Soil at Different Depths under a Long-Term Experiment Depending on Fertilization

**DOI:** 10.3390/ijerph191610460

**Published:** 2022-08-22

**Authors:** Sławomir Józef Krzebietke, Ewa Mackiewicz-Walec, Stanisław Sienkiewicz, Jadwiga Wierzbowska, Dariusz Załuski, Agata Borowik

**Affiliations:** 1Department of Agricultural and Environmental Chemistry, Faculty of Agriculture and Forestry, University of Warmia and Mazury in Olsztyn, 10-719 Olsztyn, Poland; 2Department of Agrotechnology and Agribusiness, Faculty of Agriculture and Forestry, University of Warmia and Mazury in Olsztyn, 10-719 Olsztyn, Poland; 3Department of Genetics, Plant Breeding and Biofeedstock Engineering, Faculty of Agriculture and Forestry, University of Warmia and Mazury in Olsztyn, 10-724, Olsztyn, Poland; 4Department of Soil Science and Microbiology, Faculty of Agriculture and Forestry, University of Warmia and Mazury in Olsztyn, 10-719 Olsztyn, Poland

**Keywords:** PAHs, soil layers, long-term experiment, fertilization N, P, K, Mg, Ca, manure

## Abstract

The aim of this study was to assess the effect of long-term fertilization with manure and mineral fertilizers on the content and distribution of selected polycyclic aromatic hydrocarbons (PAHs)—the content of a sum of 16 polycyclic aromatic hydrocarbons, light and heavy PAHs in two soil layers (0–30 cm and 30–60 cm). The material for the study was composed of soil samples collected from the sixth rotation in a long-term, controlled field experiment, conducted in Bałcyny since 1986. The content of 16 polycyclic aromatic hydrocarbons was determined on a gas chromatographer coupled with an FID detector. In order to evaluate the significance of differences between the mean effects on the tested characteristics, a non-parametric Mann–Whitney *U* test for two independent samples was applied. A higher content of the sum (16) of PAHs was found in the 0–30 cm than in the 30–60 cm soil layer. The research results also demonstrated a higher content of the sum of light PAHs in the 30–60 cm than in the 0–30 cm soil layer. The content of heavy PAHs, in turn, was significantly higher in the upper than in the deeper soil layer. This dependence appeared in both the soil fertilized with manure and soil nourished only with mineral fertilizers.

## 1. Introduction

Polycyclic aromatic hydrocarbons (PAHs), polynuclear aromatics (PNAs), or polycyclic organic matter (POM) are widespread organic pollutants, mainly originating from the incomplete incineration of organic matter and from some natural processes [1,2]. They are contaminants [2] that have a toxic impact on organisms, acting as carcinogens and mutagens, in addition to which they are very persistent in the soil environment [3,4,5].

The content of PAHs in soil is greater in the surface soil layers and decreases deeper into the soil profile [6,7,8]. According to Farahani et al. [9], the biodegradation of PAHs diminishes with the depth of the soil profile. Removal of PAHs from clay soils can be more difficult than from sandy ones [10]. The aging of contaminants favors the formation of stronger bonds with soil components and their diffusion into soil micropores. Polycyclic aromatic hydrocarbons are much more readily adsorbed by organic than by mineral substance [11,12].

Wang et al. [13] proved that PAHs with a smaller number of rings appeared in higher quantities in the 0–10 cm soil layers, while those with a greater molecular mass (4 to 6 rings) were more abundant in deeper layers. This can be explained by the fact that compounds with fewer rings have higher biodegradability [14]. The results obtained by Liao et al. [15], in turn, confirmed that most ‘light’ PAHs tend to be accumulated in the surface soil layers. Additionally, Cousin et al. [16] reported that PAHs accumulated in the upper soil layer and their content decreased deeper into the soil profile. The study completed by Czop and Wandrasz [17] showed that sorption of PAHs (fluorene, anthracene, pyrene) could take place in soils with a high content of organic matter in the surface layers, and that these compounds could travel within soil only due to mechanical motions or soil cracks caused by drought. In mineral soils, however, PAHs continually move downward, which is a consequence of the low content of mineral substance and good permeability of the substrate.

Long-term studies on changes in the PAH content in agricultural soils as a function of long-term fertilization in an unchanged system, both with mineral fertilizers and in combination with manure, are very scarce in the literature. The previously published findings by Krzebietke et al. [18] and Mackiewicz-Walec and Krzebietke [19] confirmed that manure was also a source of PAHs, especially in an area with a relatively low PAH contamination from air pollutants accumulating in the dust and eventually depositing on the soil surface [20]. Most studies, mainly of a monitoring nature, concern the impact of large urban agglomerations on the contamination of neighboring soils [21,22,23] or the management of waste containing PAHs. The main source of pollutant emissions in the vicinity of the research area is the municipal and residential sector followed by transport, while the least contribution comes from the point emission of pollutants. The municipal and residential sector is responsible for approximately 90% of benzo(a)pyrene (BaP) emissions. In 2018, the average annual concentration of benzo(a)pyrene in the PM10 suspended dust was 1.5 ng m^−3^ in the city of Olsztyn, approximately 45 km in a straight line from the experiment’s location, and 2 ng m^−3^ in the city of Elbląg, approximately 65 km from the experiment’s location [24].

The aim of this study was to assess the effect of long-term fertilization with manure and mineral fertilizers on the content and distribution of the selected PAHs (the content of ∑16, light and heavy PAHs) in two soil layers (0–30 cm and 30–60 cm). Long-term studies based on controlled experiments allowed us to capture the current state of pollution irrespective of changes in the climatic conditions (precipitation, temperature, wind) and plants grown under such conditions. They also provide the background for further considerations regarding the pollution of agricultural soils with PAHs.

## 2. Materials and Methods

### 2.1. Description of the Field Experiment

The research material consisted of soil samples collected in 2006–2009 from a long-term, controlled field experiment carried out in Bałcyny (Poland), 53°35′34.045″ N; 19°50′54.671″ E (Figure 1), since 1986. More detailed information about the design of the experiment and applied fertilization can be found in Krzebietke et al. [18] and Mackiewicz-Walec and Krzebietke [19] (Appendix A). The experiment was set up in three replications on a Haplic Luvisol soil [25]. According to the texture classification (USDA—United States Department of Agriculture), the soil was identified as sandy loam [19,26]. This study included soil samples taken from the sixth crop rotation cycle (sugar beet, spring barley, maize, spring wheat) [25] (Appendix A). The content of PAHs was determined in two soil layers (0–30 cm and 30–60 cm) in order to trace the vertical distribution of these compounds. Soil was sampled with a soil sampler, obtaining around 1 kg of the soil material. After being air dried, the soil was sifted through a 2 mm mesh sieve. The soil samples were stored in the air-dry state, in tightly closed dark glass bottles, at a temperature of up to 4 °C.

The content of 16 polycyclic aromatic hydrocarbons was analyzed using a Trace GC Ultra ITQ900 (Fisher Scientific THERMO, Austin, TX, USA) gas chromatographer coupled with an FID detector. The 1 h extraction of each 20 g of soil sample with 20 cm^3^ of acetonitrile was conducted using an ultrasonic cleaner and horizontal shaker. The analysis was carried out on a 30 m long Rxi-5ms column. Quality control measures included the analysis of a reference material (BNAs in Soil—R.T. Corporation, Laramie, WY, USA), duplicate matrix samples, and the solvent blank sample. The recovery of PAHs from the soil ranged from 84% to 93%, and was considered separately for each of the compounds analyzed. The detailed procedure was described in Mackiewicz-Walec and Krzebietke [19]. The sum of light PAHs was composed of naphthalene, acenaphthene, acenaphthylene, fluorene, anthracene, phenanthrene, fluoranthene, pyrene, chrysene, and the heavy ones consisted of benzo(a)anthracene, benzo(a)pyrene, benzo(b)fluoranthene, benzo(k)fluoranthene, benzo(g,h,i)perylene, indeno(1,2,3-cd)pyrene, dibenzo(a,h)anthracene.

### 2.2. Statistical Calculations

In order to verify the significance of differences between the mean effects of the analyzed variables (sum of 16, sum of light and sum of heavy PAHs) in the two soil layers of 0–30 cm and 30–60 cm, the data were checked for normality with the Shapiro–Wilk test. It confirmed that the data were not normally distributed, so a non-parametric Mann–Whitney U test for two independent samples was applied. The analyses were performed on the non-transformed data.

## 3. Results

The research was performed on soil samples collected in 2006–2009, that is, spanning one full crop rotation cycle. Generally, a significantly higher content of the sum (16) of PAHs was determined in the 0–30 cm soil layer than in the 30–60 cm one (Figure 2A). Regardless of the two different fertilization regimes (i.e., with manure or with mineral fertilizers alone), the content of the sum (16) of PAHs in soil was higher in the 0–30 cm than in the 30–60 cm soil layer (Figure 2B). Generally, a lower PAH content was recorded in the 30–60 cm soil layer (Appendix A). The 30–60 cm soil layer had a higher average content of 2- and 4-ring PAHs than the 0–30 cm soil layer, while more 3-, 5-, and 6-ring PAHs were found in the surface layer of the soil.

No significant differences were observed in the content of the sum (16) of PAHs between the two soil layers: 0–30 and 30–60 cm deep, in soil without natural fertilizer, fertilized with the smallest doses of nitrogen and potassium or in limed soil (Figure 3). T results clearly demonstrated that a higher content of PAHs, with differences confirmed statistically as significant or insignificant ones, appeared in the soil fertilized with manure.

The research results demonstrated a generally higher content of the sum of light polycyclic aromatic hydrocarbons (naphthalene, acenaphthene, acenaphthylene, fluorene, anthracene, phenanthrene, fluoranthene, pyrene, chrysene) in the 30–60 cm than in the 0–30 cm soil layer (Figure 4A). Similarly, a higher content of light PAHs in the 30–60 cm soil layer was determined in both types of soil (i.e., in soil treated with manure and in soil without manure application) (Figure 4B). However, the differences between the two soil layers proved to be statistically insignificant.

The comparison of the content of light PAHs in the two soil layers (0–30 and 30–60 cm) in each variant of fertilization proved the lack of significance of the differences (Figure 5). However, a higher content tended to be determined in the lower than in the upper soil layer.

The content of the total heavy PAHs in the upper soil layer was significantly higher than in the deeper layer (Figure 6A). This relationship appeared in both the soil treated with manure and in soil receiving only mineral fertilization (Figure 6B).

Based on the results of the statistical computations, it was demonstrated that the content of heavy PAHs in the upper soil layer was significantly higher than in the underlying, deeper soil layer, irrespective of the applied mineral fertilization (Figure 7). More detailed calculations carried out for the objects fertilized with manure or only with mineral fertilizers confirmed the above relationship, as the content of heavy PAHs was invariably higher in the upper than in the deeper soil depths, although the difference was not always verified as statistically significant in the following objects: N_0_P_0_K_0_, N_2_P_1_K_3_, N_2_P_1_K_2_Mg fertilized with manure and N_2_P_1_K_2_ with exclusive mineral fertilization.

## 4. Discussion

The study concerned the period of 2006–2009, that is, spanned one full crop rotation cycle covering all four growing seasons. The long-term fertilizer inputs to the soil contributed to an increased PAH pollution over the long-term. The results confirmed that higher content of the sum (16) of PAHs appeared in the soil fertilized with manure. Such a statement was also found in other twelve-year research [18,19]. Similar findings were reported by Mazur et al. [27], who stated that manure-fertilized soil contained the highest PAH content compared with the minerally fertilized and mineral-organically fertilized soils. The anthropogenic sources of PAHs include the products of the incomplete combustion of different organic materials [28] and the references therein [29], and their content is dependent on the type of materials and atmospheric conditions. The manure affects an increased source of PAHs in soil compared to the mineral fertilizers.

A high amount of the sum (16) of PAHs was usually recorded in the upper soil layers, which is characteristic of urban agglomerations or the neighboring areas [21,22,23]. Concerning the vertical distribution, a higher content of the sum (16) of PAHs in the 0–30 cm soil layer than in 30–60 cm one was confirmed in this study, similarly to other findings (e.g., [6,7,8,13,16,17]). Significant differences in the content of the total PAHs in the upper soil layers versus the deeper one regardless of the type of fertilization (i.e., with manure or only with mineral fertilizers) were also similar to the findings of Maliszewska-Kordybach et al. [30] and Klimkowicz-Pawlas et al. [31]. According to Baran and Oleszczuk [32], the higher PAH content in the upper soil layers results from their large affinity for organic matter. Czop and Wandrasz [17] maintained that PAH compounds bind to organic matter in the topmost soil layers of organic and cultivated soils, and the translocation of these compounds can only take place in response to mechanical movements or cracks caused by drought.

Slightly different results concerning the content of light and heavy PAHs in soil are reported by Mazur et al. [27] and Zhang et al. [33]. These authors claimed that light PAHs that prevailed over heavy PAHs in the upper soil layers could suggest a lower anthropogenic pressure from agriculture. The prevalence of heavy PAHs in the upper soil layer, in turn, could indicate an enhanced anthropogenic source of contamination [23,34]. More intensive PAH accumulation was found along the pollution transect from the northeastern to southwestern areas of Poland [35]. It was found that the main sources of PAHs came from the coal and wood combustion. The origin of PAHs in the soil is connected to the decomposition of airborne particles that were emitted from natural as well as anthropogenic sources [36,37].

An elevated content of the sum of light polycyclic aromatic hydrocarbons such as naphthalene, acenaphthene, acenaphthylene, fluorene, anthracene, phenanthrene, fluoranthene, pyrene, and chrysene was found in the 30–60 cm layer. The reason could be the greater water solubility of light PAHs and their ability to translocate to the deeper soil layers [38,39]. Wang et al. [13], in turn, confirmed that heavy PAHs are rather difficult to translocate into the soil vertical profile. The higher content of heavy PAHs in the deeper soil layers determined in this study can be explained by their greater sorption to organic matter. This was confirmed by a higher content of total organic carbon in the 0–30 cm soil layer (approximately 11.0 g kg^−1^, on average) than in the 30–60 cm one (approximately 9.0 g kg^−1^ on average) in this experiment [40]. Similar findings of higher PAH content in the upper soil layers (O-horizons) than in the deeper soil layers (A/B-horizons) were confirmed from high-altitude mountains in China [41]. A higher accumulation of PAHs relating to soil organic matter present in a higher content in the surface layers was also recorded in the case of forest stands [42]. In the present studies, however, by comparing the high content of light PAHs in the deeper soil layers, the differences between the soil treated with manure and soil without manure application were statistically insignificant.

## 5. Conclusions

A higher content of the sum (16) of heavy PAHs was determined in the 0–30 cm soil layer than in the 30–60 cm one. This relationship appeared in both the soil fertilized with manure and soil receiving only mineral fertilization. No significant differences in the total content of 16 PAHs between the two soil layers of 0–30 and 30–60 cm were detected in the soil without natural fertilizer and supplied with the lowest doses of nitrogen and potassium as well as in the limed soil.

The results distinctly confirm that higher quantities of PAHs appeared in the manure-treated soil. Heavy PAHs occurred in higher amounts in the upper soil layer (0–30 cm), while light PAHs were more abundant in the deeper soil layer (30–60 cm).

## Figures and Tables

**Figure 1 ijerph-19-10460-f001:**
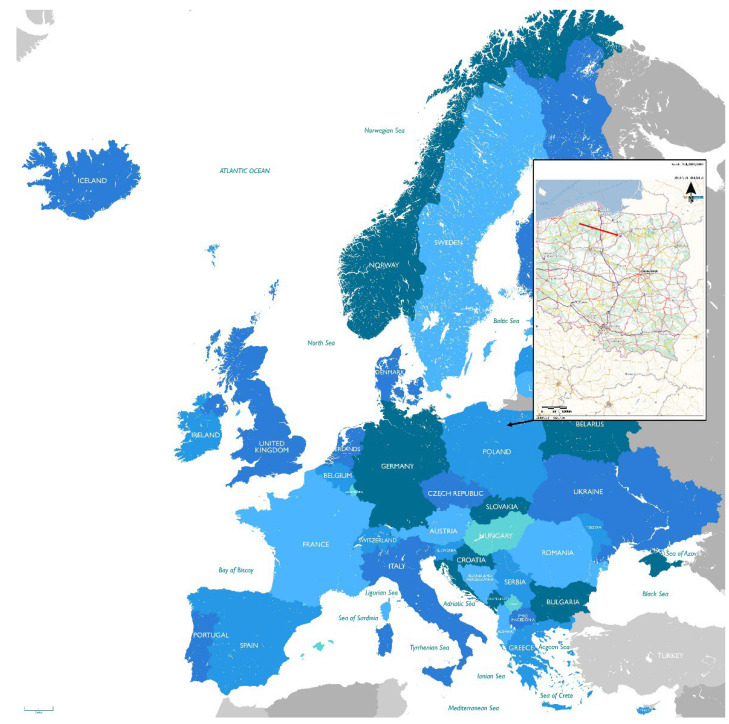
The localization of the experiment.

**Figure 2 ijerph-19-10460-f002:**
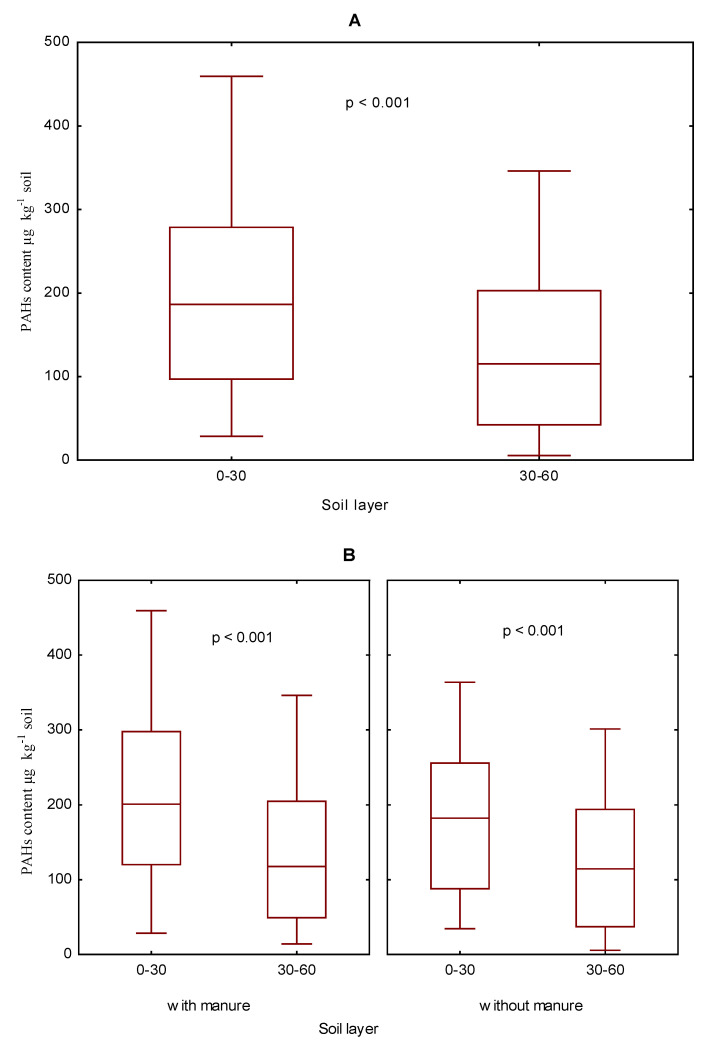
The content of the sum (16) of PAHs in the soil in the years 2006–2009 depending on the soil layer (**A**). The 030 cm and 30–60 cm and manure fertilization (**B**) in µg kg^−1^.

**Figure 3 ijerph-19-10460-f003:**
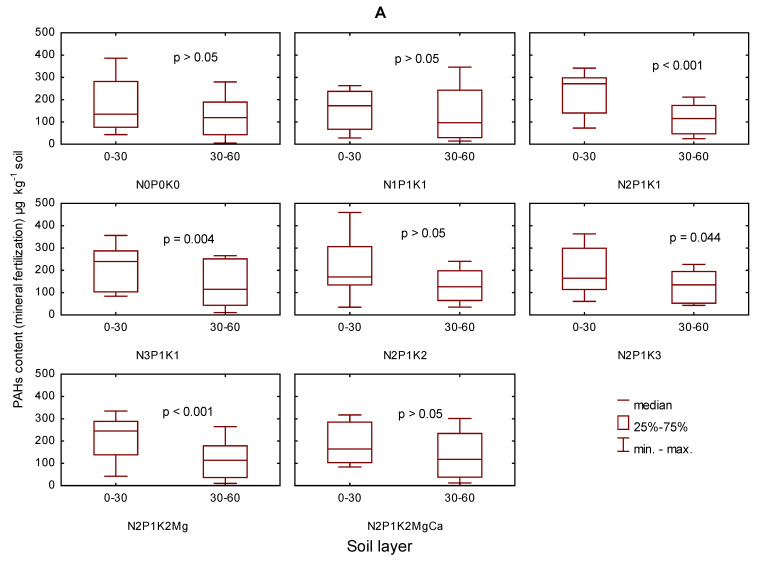
The content of the sum (PAHs) of in soil (0–30 cm and 30–60 cm) in 2006–2009 depending on mineral fertilization (**A**), and manure-mineral and mineral fertilization (**B**) in µg kg^−1^; the codes were used for variants of fertilization and are given in Appendix A.

**Figure 4 ijerph-19-10460-f004:**
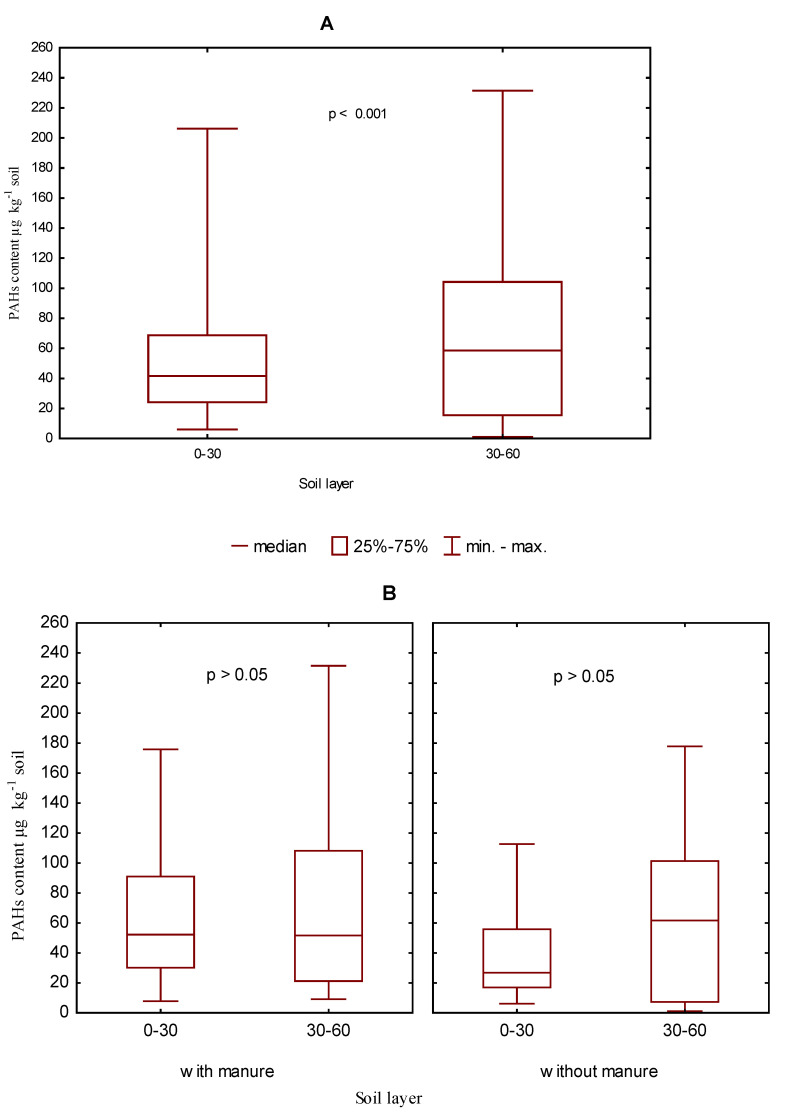
The content of the sum of light PAHs in the soil in 2006–2009 depending on the soil layer (**A**). The 0–30 and 30–60 cm and manure fertilization (**B**) in µg kg^−1^.

**Figure 5 ijerph-19-10460-f005:**
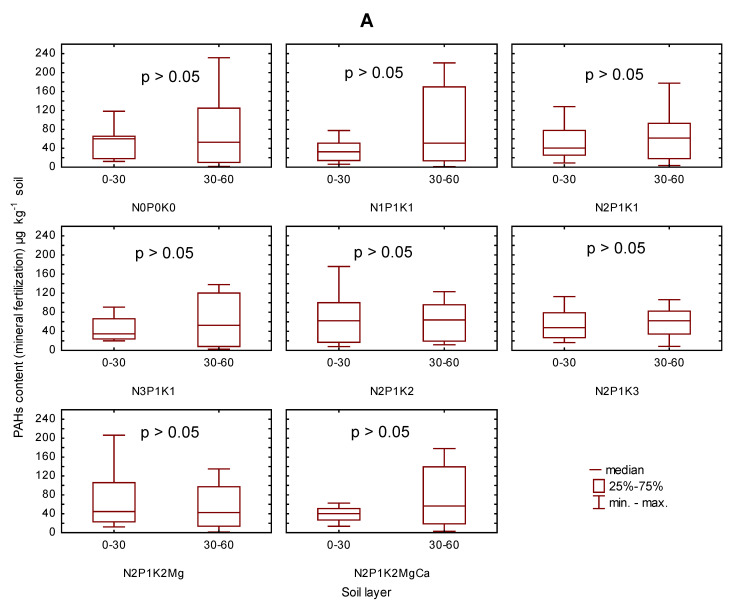
The content of the sum of the light PAHs in soil (0–30 and 30–60 cm) in 2006–2009 depending on mineral fertilization (**A**) and manure-mineral and mineral fertilization (**B**) in µg kg^−1^.

**Figure 6 ijerph-19-10460-f006:**
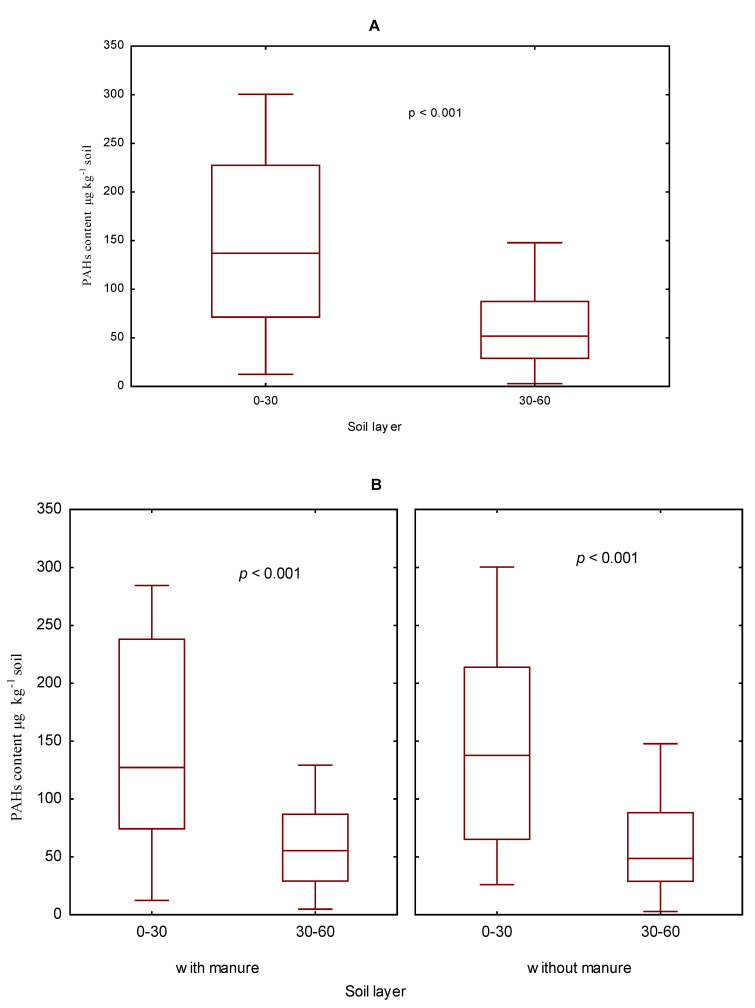
The content of the sum of heavy PAHs in soil in 2006–2009 depending on the soil layer (**A**). The 0–30 and 30–60 cm and manure fertilization (**B**) in µg kg^−1^.

**Figure 7 ijerph-19-10460-f007:**
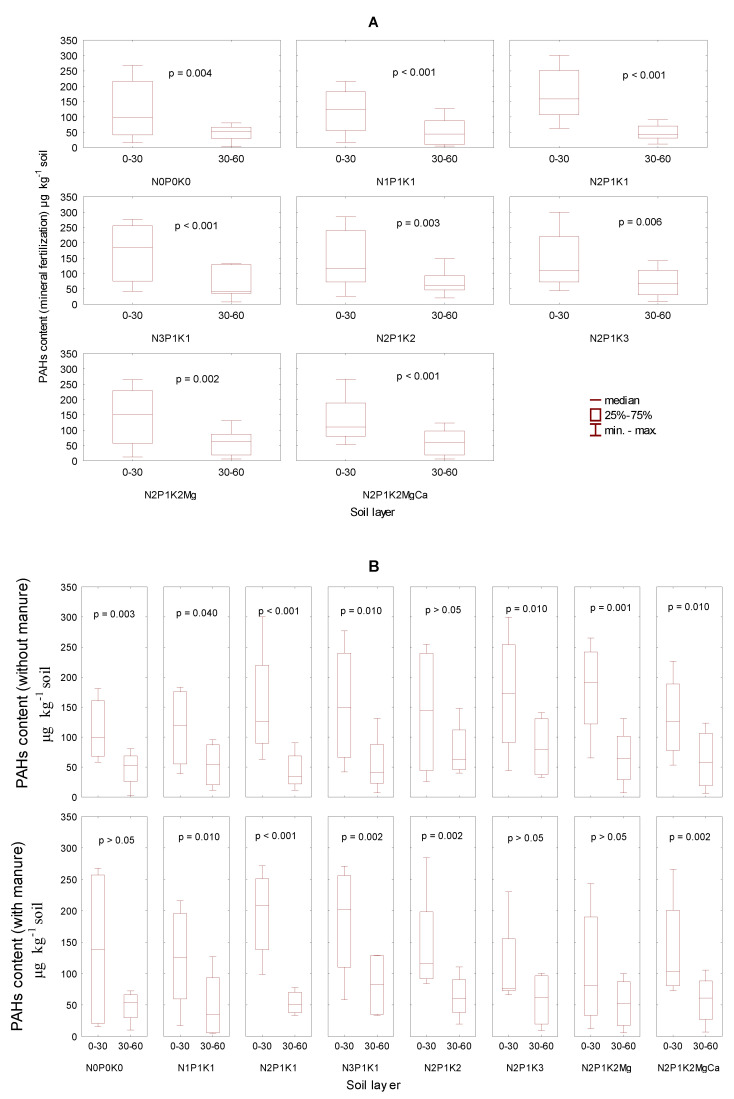
The content of the sum of heavy PAHs in soil (0–30 and30–60 cm) in 2006–2009 depending on the mineral fertilization (**A**) and manure-mineral fertilization (**B**) in µg kg^−1^.

## Data Availability

The datasets generated during and/or analyzed during the current study are available from the corresponding author on reasonable request.

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
