# Peer review of "Polycyclic Aromatic Hydrocarbons in Soil at Different Depths under a Long-Term Experiment Depending on Fertilization"

_ijerph, 2022, doi:10.3390/ijerph191610460_

Round 1

Reviewer 1 Report

The revised manuscript can be accepted for publication.

Reviewer 2 Report

This paper can be accepted now.

This manuscript is a resubmission of an earlier submission. The following is a list of the peer review reports and author responses from that submission.

Round 1

Reviewer 1 Report

In the present report, the authors studied the effect of long-term fertilization with manure and mineral fertilizers on the content and distribution of selected PAHs (the content of Σ16, light and heavy PAHs) in two soil layers (0-30 cm and 30-60 cm). It is a worthy study, especially the rarity of soil samples. However, the leak of research novelty and proper investigation make it unsuitable for the publication in our journal. The results and discussion section of the study is not in-depth, it is more like a list of results, and there is no clear explanation for these results, and the form of results in the study is too single. Therefore, I recommend to reject it and advise the following comments for improving the quality of the manuscript for further submission.

1. The references cited in the study should be updated. 

2. Details on the extraction and analysis of PAHs should be given briefly in the SI.

3. Please add the description of quality assurance and quality control in the manuscript.

4. The titles of horizontal and vertical axis in figures are not normalize.

5. The section of results and discussion was too simple, lack of in-depth discussion, such as the relationship between the concentration of pollutants in soil and the composition of different fertilizers.

Reviewer 2 Report

The work dealt with the detection of PAH in soils, compared with upper soil in 0-30cm depth and lower 30-60cm in depth. I was wondering why using sum of the PAHs instead of individual.

1. GC-FID analysis should be able to give a wealth of individual PAH information, rather than its totals. Is there any difference between in 0-30cm and 30-60cm in depth by the analysis of individual?

2. It seems that it has not discussed in details where the PAHs came from, containing in fertilizers themselves or other sources?

3. Why the studied area chosen? Could the geographic location map be provided in SI?

Reviewer 3 Report

Although the authors trying to present a relevant research topic of this Journal, I have some questions and comments on the current version of the manuscript. Importantly, they have used old data, so I wonder if the same data have been published previously in the same context. If they have used the same data in a different context, it would be fine, but I suggest them to provide us the access for such publication/s. Moreover, the figures and tables need appropriate formatting to meet the publishable quality. Additional comments for the improvements are given in the attached manuscript. I recommend minor revisions to this paper so that the authors can have an opportunity to greatly improve their work that can be useful and readable. Thank you.
